# A Method of Noise Reduction for Radio Communication Signal Based on RaGAN

**DOI:** 10.3390/s23010475

**Published:** 2023-01-01

**Authors:** Liang Peng, Shengliang Fang, Youchen Fan, Mengtao Wang, Zhao Ma

**Affiliations:** School of Space Information, Space Engineering University, Beijing 101416, China

**Keywords:** radio communication signal, noise reduction, RaGAN, Bi-LSTM, deep learning, modulation recognition

## Abstract

Radio signals are polluted by noise in the process of channel transmission, which will lead to signal distortion. Noise reduction of radio signals is an effective means to eliminate the impact of noise. Using deep learning (DL) to denoise signals can reduce the dependence on artificial domain knowledge, while traditional signal-processing-based denoising methods often require knowledge of the artificial domain. Aiming at the problem of noise reduction of radio communication signals, a radio communication signal denoising method based on the relativistic average generative adversarial networks (RaGAN) is proposed in this paper. This method combines the bidirectional long short-term memory (Bi-LSTM) model, which is good at processing time-series data with RaGAN, and uses the weighted loss function to construct a noise reduction model suitable for radio communication signals, which realizes the end-to-end denoising of radio signals. The experimental results show that, compared with the existing methods, the proposed algorithm has significantly improved the noise reduction effect. In the case of a low signal-to-noise ratio (SNR), the signal modulation recognition accuracy is improved by about 10% after noise reduction.

## 1. Introduction

As the premise of signal demodulation, radio signal modulation recognition is an important research content in the field of communication. It plays an important role in military and civil communications, and is of great significance to future 6G communications [1]. However, due to the complexity of the signal transmission environment and the problems of the receiving equipment, the received signal always has a certain amount of noise, which brings difficulties to the modulation recognition of the received signal. In recent years, modulation recognition algorithms have developed rapidly, but there is a common problem that the recognition accuracy is not ideal in low SNR scenarios [2]. The noise reduction of the received signal is an effective means to solve this problem. Therefore, it is of great significance to obtain a noise reduction algorithm that can reduce the signal noise while maintaining the basic characteristics of the signal for signal modulation recognition.

Signal denoising is to extract clean signals from noisy signals, which has always been a problem in the field of signal processing. Effective signal denoising is of great significance for radio signal communication. The existing signal denoising methods are mostly based on signal processing methods, such as wavelet transform (WT)-based denoising methods [3,4], empirical mode decomposition (EMD)-based denoising methods [5], etc. The denoising effect based on WT is closely related to the number of decomposition layers, wavelet basis function, threshold selection, etc. Too few decomposition layers affect the denoising effect, and too many decomposition layers will lead to signal distortion, while the selection of the wavelet basis function and the optimal threshold is usually determined by experience. Although the noise reduction algorithm based on EMD has advantages in dealing with nonlinear and nonstationary signals, there are problems such as mode aliasing and end effect.

However, DL can extract the hidden feature values from data relatively accurately and conveniently, and has efficient data processing capability. In recent years, DL technology has been widely used in speech processing [6], image processing [7], natural language processing [8], and other fields. With the development of technology, DL technology is also used in the field of signal processing. In addition, signal denoising has always been a difficult problem in the field of signal processing, so some scholars have tried to use DL to solve this problem. A detailed description of the signal noise reduction method based on DL will be discussed later in the second section. Although these works have achieved good results, there are still some drawbacks: (i) The perfect separation of signal and noise cannot be realized [9,10,11]; (ii) The existing noise reduction methods based on the generative adversarial network (GAN) [12] still have the problems of long training time and slow convergence speed in the training process [13,14]; (iii) The requirements for the number of signal sampling points are high. When the number of signal sampling points is too small and the waveform is not obvious, a satisfactory denoising effect cannot be achieved [15].

To solve the above problems, this paper proposes a RaGAN-based radio signal noise reduction method. This method takes the time-domain waveform of the radio-received signal as the processing object, uses RaGAN [16] to replace the original GAN to accelerate the convergence speed of the model, uses Bi-LSTM [17] as the core to build a generator and discriminator, effectively extracts the time-dimension characteristics of the signal, and retains the essential feature information of the signal after noise reduction. Compared with the existing methods, this method has a satisfactory denoising effect and requires fewer sampling points of target signals. In addition, we use the CNN2 [18], IQCNet [19], and IQCLNet [19] classification networks for the modulation recognition of noise-reduced signals, effectively improving the recognition accuracy of low SNR signals by about 10%, which is of great significance for signal modulation recognition.

To summarize, the main contributions of our work are:A noise reduction algorithm based on RaGAN is proposed for time-series data of radio communication signals;Our method preserves the essential characteristics of the signal after noise reduction of the radio signal;The experimental results show that the accuracy of modulation recognition is improved by about 10% after using this method to denoise the signal with low SNR compared with the signal before noise reduction.

The remainder of this paper is organized as follows. The second section describes the related work; the third section introduces the background, including the definition of the problem, the system model, and the theory of GAN; in the fourth section, the noise reduction model, dataset, network structure, and loss function are described; the fifth section shows the experimental process and discusses the results; finally, the paper is summarized in the sixth section, and further work is pointed out. Table 1 provides the abbreviations that appear in the paper.

## 2. Related Work

In the past few years, research based on DL has become an active topic in the field of signal processing [20]. Many scholars have proposed some signal denoising methods based on DL. These methods use a DL framework to extract signal features for signal denoising in an end-to-end manner. Wang et al. preprocessed a signal with impulse noise before signal modulation recognition [21]. In the case of fewer labeled samples, the modulation recognition accuracy of the underwater acoustic signal was improved by more effective extraction of signal features through denoising and task driving. However, this method is only suitable for the underwater impulsive noise environment, and is not suitable for the additive white Gaussian noise (AWGN) channel. In References [13,14], the authors used the convolutional neural network (CNN) as the core to build a generation countermeasure network to denoise electrocardiogram (ECG) signals. Although the least-squares generative adversarial network (LSGAN) [22], the conditional generative adversarial network (CGAN) [23], and other methods were used to improve the model and achieved a better denoising effect than traditional methods, there are still problems such as slow model convergence. In References [9,10,11], the authors transformed the problem of signal denoising into that of image denoising, processed the image with CNN, and achieved good results in the field of seismic signal and transient electromagnetic (TEM) signal denoising. This method will not cause large deviation in amplitude, time, and phase information after signal denoising, but it cannot achieve perfect separation of signal and noise. Soltani et al. [15] used the artificial neural network (ANN) to denoise the measured radio frequency (RF) signal sent by the local discharge source, and converted the denoising problem into a curve-fitting problem. However, this method can achieve the denoising effect only when there are enough sampling points and the waveform is obvious, while when there are too few sampling points and the waveform is not obvious, it cannot achieve a good denoising effect.

## 3. Background

In this section, we describe the problem definition and system model. In addition, since the method we use is based on GAN, we also briefly explain its basic principle and an improved training method, namely RaGAN.

### 3.1. Problem Definition and System Model

The signal in the wireless channel is transmitted through the propagation of electromagnetic waves in space, while there are some unwanted signals in the channel, which are collectively called noise. Noise is a kind of interference in the channel. Since the noise is superimposed on the signal, it is also called additive interference. Noise will have adverse effects on signal transmission. It not only limits the transmission rate of information, but also leads to signal distortion.

Gaussian white noise reflects the additive noise in the actual channel, and can nearly represent the characteristics of channel noise [24]. Therefore, the proposed method is applied to the AWGN channel. In the AWGN channel, the output signal y is expressed as the superposition of input signal m and Gaussian noise signal n:(1)y=m+n
where $n∼N\left(\mu ,\sigma^{2}\right)$. The purpose of denoising is to filter the noise from the signal polluted by noise as much as possible (i.e., denoised signal), $\hat{y}$, so as to minimize the expected *error* between the input signal and the denoised signal:(2)$$error=E{\left\|\hat{y}-y\right\|}_{2}^{2}$$
where E represents the expectation operator.

### 3.2. Generative Adversarial Network

GAN was put forward by Goodfellow et al. in 2014 [12] and soon became a research hotspot. GAN mainly includes two parts, discriminator (*D*) and generator (*G*), which solve the minimum and maximum problem of antagonism through alternate training. The idea of this model is that *D* tries to distinguish whether the input data are real data or data generated by *G*, and the distribution of data generated by *G* $p_{G}\left(x\right)$ is as close to the real data distribution as possible $p_{data}\left(x\right)$. In this way, *G* can learn to create solutions that are highly close to real data, and it is difficult to distinguish them by *D*. The objective optimization function of GAN is as follows:
(3)$$\underset{G}{\min}\underset{D}{\max}V\left(D,G\right)=E_{x∼p_{data}\left(x\right)}\left[\log\left(D\left(x\right)\right)\right]\;+E_{z∼p_{z}\left(z\right)}\left[\log\left(1-D\left(G\left(z\right)\right)\right)\right]\;$$

In the above formula, z is the input following the random distribution $p_{z}\left(z\right)$, $V\left(G,D\right)$ is the combined loss function of GAN, $G\left(z\right)$ represents the data generated by *G*, and $D\left(x\right)$ and $D\left(G\left(z\right)\right)$ represent the probability that *D* gives correct discrimination to real data and generated data.

Although GAN has achieved great success, there are some problems in training stability, and the emergence of RaGAN has solved this problem well. The *D* of standard GAN will estimate the probability of whether an input data is generated data $x_{f}$ or real data $x_{r}$, while the relative discriminator of RaGAN will average the probability that the real data $x_{r}$ is more real than the generated data $x_{f}$, so that *G* can generate higher-quality samples and accelerate the model convergence. We define the relative *D* as $D_{RaGAN}$, as shown below:(4)$$D_{RaGAN}\left(x_{r},x_{f}\right)=\sigma \left(C\left(x_{r}\right)-E\left[C\left(x_{f}\right)\right]\right)$$

In the above formula, $x_{f}=G\left(z\right)$, σ represents sigmoid function, and $C\left(s\right)$ represents the output of the discriminator without transformation, i.e., $D\left(x\right)=\sigma \left(C\left(x\right)\right)$. The loss functions $L_{G}^{RaGAN}$ and $L_{D}^{RaGAN}$ of RaGAN *G* and *D* are, respectively, expressed as:(5)$$L_{G}^{RaGAN}=-E\left[\log\left(1-D_{RaGAN}\left(x_{r},x_{f}\right)\right)\right]-E\left[\log\left(D_{RaGAN}\left(x_{f},x_{r}\right)\right)\right]$$
(6)$$L_{D}^{RaGAN}=-E\left[\log\left(D_{RaGAN}\left(x_{r},x_{f}\right)\right)\right]-E\left[\log\left(1-D_{RaGAN}\left(x_{f},x_{r}\right)\right)\right]$$

## 4. Signal Denoising Method Based on RaGAN

In this section, the noise reduction model and dataset details used are described, and the design of the network structure is given in detail. Finally, the design of the loss function is given.

### 4.1. Denoising Model

The noise in the wireless channel can interfere with the radio signal transmission. By reducing the noise component in the received signal, the original useful signal can be further restored. Using traditional signal denoising methods to separate signal from noise usually requires a lot of artificial knowledge. Noise reduction through deep DL technology can reduce the dependence on artificial domain knowledge. Therefore, a noise reduction algorithm based on RaGAN for radio communication signals is proposed in this paper. The noise reduction model and its training and testing process are shown in Figure 1.

First, the model is trained, and the signal of superimposed noise is taken as the input of the generator. The generator generates the signal by extracting the characteristics of the input signal. Then, the generated signal and the corresponding clean signal are input into the discriminator in turn. The discriminator judges the input signal and determines whether the input signal is generated by the generator or a clean signal without noise. After iterative training, the error backpropagation algorithm is used to optimize the model itself. The final discriminator cannot determine whether the input signal is the signal generated by the generator or the clean signal without noise. After the model training is completed, the generator has the ability to map the input noisy signal to the corresponding clean signal. At this time, the generator can be used to restore the noisy signal to a clean signal. This is the principle of RaGAN noise reduction in this paper.

### 4.2. Dataset

This paper adopts the public dataset RML2016.10a [25]. Since this dataset was originally used for signal modulation recognition, in order to make this dataset more suitable for signal noise reduction training, signals with SNR of 18 dB of eight modulation methods in the RML2016.10a dataset were selected for wavelet denoising and smoothing to form clean signal samples. We randomly add Gaussian white noise with different SNRs to the clean signal samples, and finally generate the corresponding noisy signal samples. In practice, if the signal SNR is too high, it will be too ideal, while if the SNR is too low, the characteristics of the signal will be completely covered by noise. Therefore, the SNR of the noisy signal is between −8 dB and 10 dB, with an interval of 2 dB. The visual image of the signal is shown in Figure 2. There are 80,000 signal samples in total. The details of the dataset are shown in Table 2.

### 4.3. Network Structure

Extracting time-dimension information is a key step in radio signal processing. Bi-LSTM [17] is composed of forward long short-term memory (LSTM) [26] and backward LSTM, which makes up for the defect that one-way LSTM cannot encode backward to forward information. It can effectively extract the time-dimension features of signals, and has a good performance in processing time-series data. So, we build generators and discriminators with Bi-LSTM as the core.

#### 4.3.1. Generator Network Structure

As shown in Figure 3, the generator is composed of two full connection layers and two layers of Bi-LSTM. The input and output dimensions of the generator are 128. The input dimension of the Bi-LSTM layer is 1, the number of hidden layer nodes is 128, and the output dimension is 128. After the full connection layer output, Dropout [27] and Leaky ReLU are used to activate the output nonlinearly. Therefore, after the generator inputs the signal, it performs layer normalization (LN) [28], and then extracts the waveform features and time-dimension features of the noisy signal through full connection mapping and the Bi-LSTM layer to achieve the purpose of signal denoising. Before outputting the data, it performs normalization restoration to ensure the generalization ability of the model, and finally outputs the denoised signal.

#### 4.3.2. Discriminator Network Structure

The discriminator network structure is shown in Figure 4, which is also composed of two layers of a full connection layer and two layers of Bi-LSTM. The discriminator input dimension is 128, and the output dimension is 1. The input dimension in the Bi-LSTM layer is 1, the number of hidden layer nodes is 128, and the output dimension is 128. After the full connection layer is output, Leaky ReLU and Dropout are introduced to prevent the model from overfitting. Therefore, after the discriminator inputs the signal data, it performs LN on the input, extracts the difference between the real signal data and the waveform in the generated signal data through the Bi-LSTM layer, and then outputs the decision result through the mapping of the full connection layer.

### 4.4. Loss Function

The definition of the loss function is very important for model performance. In this paper, the relative discriminator is used. Compared with the standard discriminator, which will judge whether the input signal is clean signal $s_{r}$ or generated signal $s_{f}$, the relative discriminator will estimate the relatively more true probability of clean signal $s_{r}$ than generated signal $s_{f}$. The loss function of discriminator $L_{D}^{Ra}$ is defined as follows:(7)$$L_{D}^{Ra}=-E\left[\log\left(D_{RaGAN}\left(s_{r},s_{f}\right)\right)\right]-E\left[\log\left(1-D_{RaGAN}\left(s_{f},s_{r}\right)\right)\right]$$

The counter loss function of the generator $L_{G}^{Ra}$ is expressed as:(8)$$L_{G}^{Ra}=-E\left[\log\left(1-D_{RaGAN}\left(s_{r},s_{f}\right)\right)\right]-E\left[\log\left(D_{RaGAN}\left(s_{f},s_{r}\right)\right)\right]$$
where $s_{f}=G\left(s\right)$. By using a relative discriminator to accelerate the model convergence, the generator can produce a higher-quality denoising signal.

Inspired by [29,30], we express the overall loss function of the generator as the weighted sum of content loss and confrontation loss, so that the generator is affected by both content loss and confrontation loss, as shown below:(9)$$L_{G}=L_{con}+\lambda L_{G}^{Ra}$$
where $L_{con}$ represents the content loss, $L_{G}^{Ra}$ represents the confrontation loss, and λ is the coefficient to balance different loss items. Through the common constraint of content loss and confrontation loss, the generator can better restore noisy signals s to clean signals $s_{r}$. $L_{MSE}$ and $L_{1}$ are used to construct the content loss function.
(10)$$L_{MSE}=\frac{1}{n}\sum_{i=1}^{n}{\left(s_{r_{i}}-G{\left(s\right)}_{i}\right)}^{2}$$
(11)$$L_{1}=\frac{1}{n}\sum_{i=1}^{n}\left|s_{r_{i}}-G{\left(s\right)}_{i}\right|$$

In the above formula, $L_{MSE}$ represents the mean square error between the denoised signal $G\left(s\right)$ and the clean signal $s_{r}$, and $L_{1}$ represents the absolute value error between the denoised signal $G\left(s\right)$ and the clean signal $s_{r}$, where $G\left(s\right)$ is the output of the generator and n is the length of a single sample. The content loss function $L_{con}$ is expressed as:(12)$$L_{con}=\left(L_{MSE}+L_{1}\right)/2$$

## 5. Experiments

In this section, the method proposed in this paper is applied to wireless signal noise reduction, and the training details and parameters are described. In order to verify the effectiveness of the proposed method, we compare it with the existing algorithm and make a comparative analysis. Finally, a Bi-LSTM analysis experiment was conducted. The hardware and software environment of the experiment are shown in Table 3.

### 5.1. Training Details and Parameters

We divide the dataset into a training set and a test set according to 4:1. The training set contains 64,000 signal data, and the test set contains 16,000 signal data. Because there are few hyperparameters, we use grid search to select hyperparameters. First, define the traversal interval λ = {0.5, 0.05, 0.005, 0.0005}, batch size = {256, 512, 1024, 2048}, learning rate = {0.0001, 0.001,0.01, 0.1}, and optimizer = {Adam}, and then calculate the cost function of all hyperparametric combinations on the validation set to obtain the optimal hyperparametric set in the interval. The epoch is determined by observing the convergence of the loss function. The final hyperparameters are shown in Table 4:

The initial learning rate of the generator and discriminator is set to 0.001, and the learning rate is halved after 50 times of training. During the training process, the model is iteratively optimized by calculating the loss function. After the model training is completed, the final weight model of the generator and discriminator is saved.

### 5.2. Experimental Result

In order to verify the noise reduction performance of this method, the test set is tested on the trained generator, and the signal noise reduction performance is compared with WT, EMD, and standard GAN noise reduction models under different input signal-to-noise ratios. The output SNR is used as the evaluation standard of noise reduction performance.
(13)$$\mathrm{SNR}=10\log_{10}\left(\frac{P_{s}}{P_{n}}\right)$$
where $P_{s}$ and $P_{n}$ are the effective power of signal and noise, respectively. The input signal-to-noise ratio of the noisy signal test set is within the range of [−8, 10] dB, and the step size is 2 dB. The comparison diagram of time-domain waveforms of 8PSK signals before and after noise reduction by different methods is shown in Figure 5.

It can be seen from Figure 6 and Table 5 that the noise reduction performance of this method is significantly improved compared with the two traditional methods. The SNR is improved by about 10 dB for signals with an input of −8 dB to 0 dB, and the SNR increases by about 8 dB for signals with an input of 0 dB to 10 dB. The noise reduction performance is also improved compared with the standard GAN when the input signal is more than −2 dB. Compared with the WT and EMD noise reduction methods, the output signal-to-noise ratio is improved by about 4dB under the condition of 0 dB signal-to-noise ratio. This shows that the method has significantly improved the performance of signal noise reduction.

In order to verify that the model can retain the original features of the signal after signal noise reduction, we use the RaGAN noise reduction model to reduce the noise of the signal in the original RML2016.10a data set, and use CNN2, IQCNet and IQCLNet classification networks to classify the modulation recognition of the signal before and after noise reduction. By comparing the recognition rate of the signal modulation classification before and after noise reduction with the RaGAN noise reduction algorithm, we can judge the degree of the original features of the signal after noise reduction. The hyperparameter of the classification network model is shown in Table 6.

It can be seen from Figure 7 that the accuracy of signal modulation recognition is effectively improved after noise reduction of the signal through this model, especially in the low SNR range. In (a), the CNN2 classification network is used for the modulation recognition of signals. The recognition accuracy is improved by nearly 10% when the input signal SNR is between −14 dB and −20 dB. In (b), the IQCNet classification network is used for signal modulation recognition. The recognition accuracy is improved by nearly 10% when the input signal SNR is between −4 dB and −12 dB, and by nearly 5% when the SNR is between −12 dB and −20 dB. In (c), the IQCLNet classification network is used for the modulation recognition of signals. The recognition accuracy is improved by nearly 10% when the input signal SNR is between −8 dB and −14 dB, and by nearly 6% when the SNR is between −14 dB and −20 dB. However, in the range of high SNR, since the signal features are obvious, the feature prominence effect of a high SNR signal after noise reduction is not as good as that of the low SNR signal, so the improvement of the high SNR signal recognition accuracy is not high.

Because the algorithm is used for communication applications, the complexity of a large number of calculations and algorithms should also be considered. A total of 1024 noisy signals are processed by the algorithm, WT, and EMD in this paper, and the processing time comparison is shown in Table 7. It can be seen from Table 7 that when processing the same number of noisy signals, the algorithm proposed in this paper takes the least time, which is more conducive to the application of the algorithm in practice.

### 5.3. Exploration of Bi-LSTM in the Model

During the experiment, we found that the number of hidden layer nodes used in the Bi-LSTM layer will affect the noise reduction effect of the model. In order to find the optimal number of solution points, we made an experimental comparison of the number of hidden layer nodes used by Bi-LSTM.

It can be seen from Figure 8 and Table 8 that the noise reduction performance of the model is closely related to the number of hidden layer nodes in Bi-LSTM. With the increase in the number of nodes used, the noise reduction performance of the model is also improved. When the number of hidden layer nodes is 128, the noise reduction performance of the model is the highest. However, when the number of hidden layer nodes exceeds 128, due to the large number of model parameters, the performance of the model cannot be improved by increasing the number of nodes.

## 6. Conclusions

Aiming at the degradation of radio communication signal reception quality, this paper proposes a RaGAN-based radio communication signal noise reduction model. This model uses Bi-LSTM as the core to build a generator and discriminator. Through Bi-LSTM, signal features are effectively extracted, the time dependence of data is learned, and the weighted loss function is used to train the model to achieve end-to-end denoising of radio signals. Compared with several signal noise reduction methods, this method has significantly improved the signal noise reduction performance, and has been verified on the public dataset RML2016.10a, which proves the effectiveness of this model.

However, in this experiment, we only processed the signal from the perspective of the time domain, which has certain limitations. Therefore, it is the work we need to conduct in the future to use deep learning technology combined with signal frequency domain and other aspects to process signals. In addition, our method is mainly applicable to signal propagation in the AWGN channel. In future research, we will apply the proposed method to Rayleigh fading to further verify the feasibility of this method, so as to improve the generalization ability of the model.

## Figures and Tables

**Figure 1 sensors-23-00475-f001:**
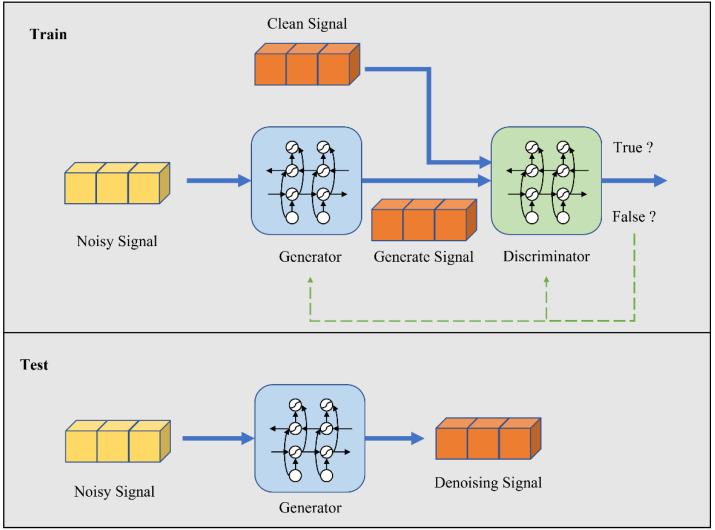
Signal noise reduction model based on RaGAN and training and testing process.

**Figure 2 sensors-23-00475-f002:**
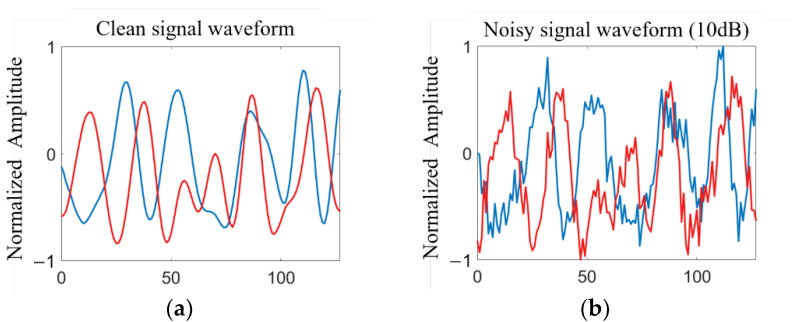
Radio signal dataset: (**a**) original clean signal waveform; (**b**) noisy signal waveform.

**Figure 3 sensors-23-00475-f003:**
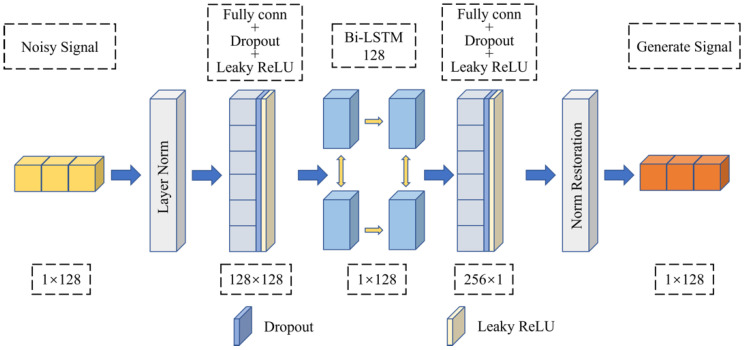
Generator network structure.

**Figure 4 sensors-23-00475-f004:**
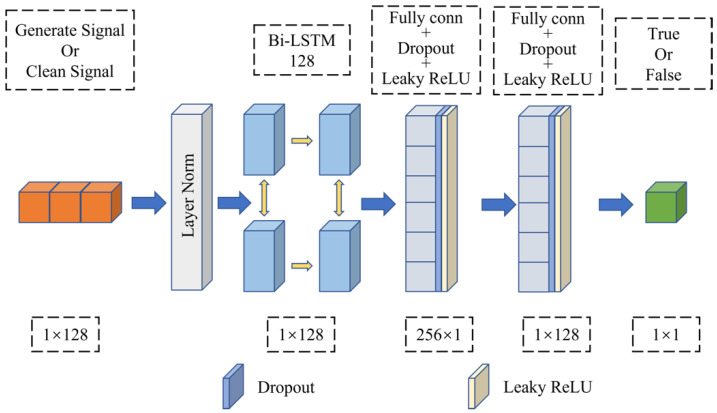
Discriminator network structure.

**Figure 5 sensors-23-00475-f005:**
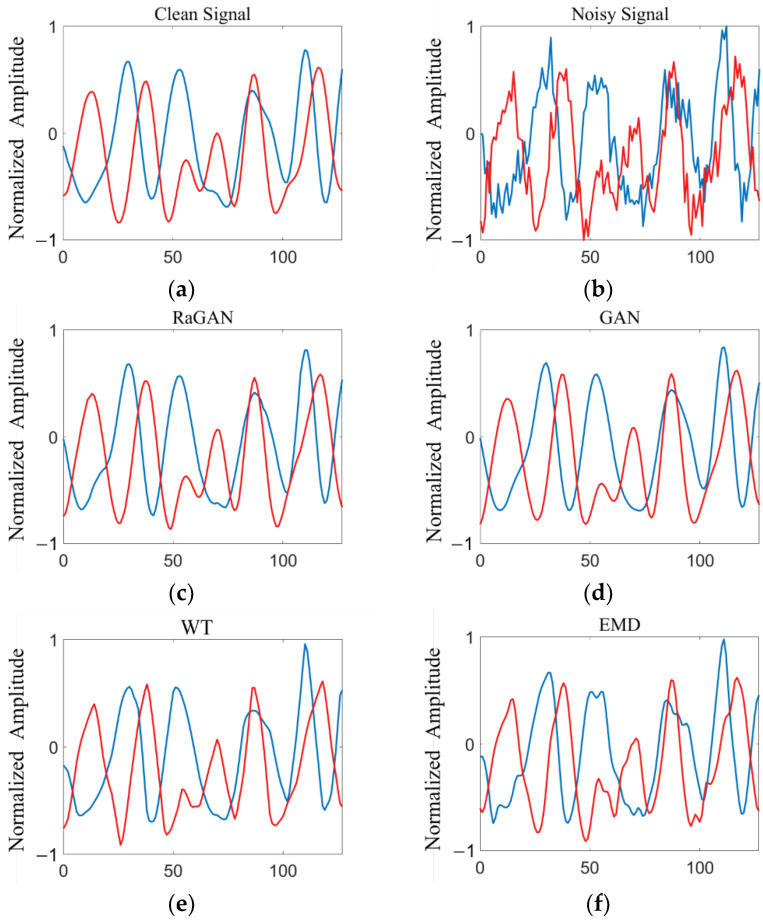
Time-domain waveform comparison of 8PSK signal before and after noise reduction: (**a**) original clean signal waveform; (**b**) noisy signal waveform; (**c**) time-domain waveform of signal after RaGAN noise reduction; (**d**) time-domain waveform of signal after GAN noise reduction; (**e**) time-domain waveform of signal after WT noise reduction; (**f**) time-domain waveform of signal after EMD noise reduction.

**Figure 6 sensors-23-00475-f006:**
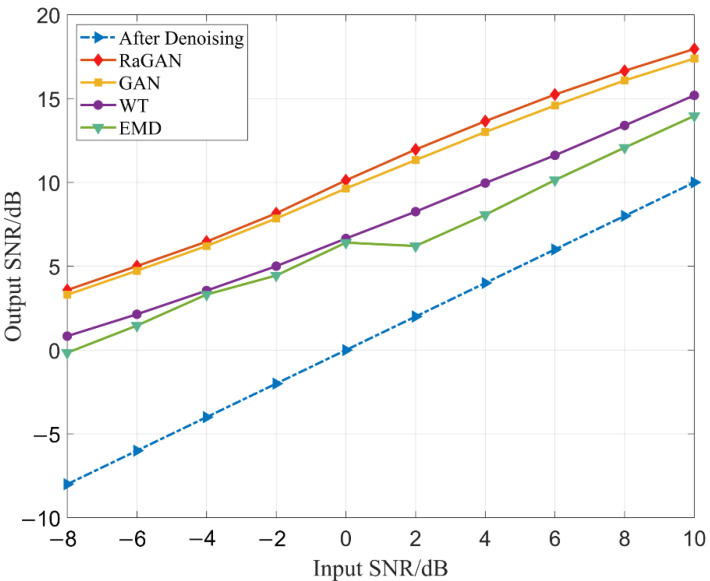
Comparison of noise reduction performance of different methods.

**Figure 7 sensors-23-00475-f007:**
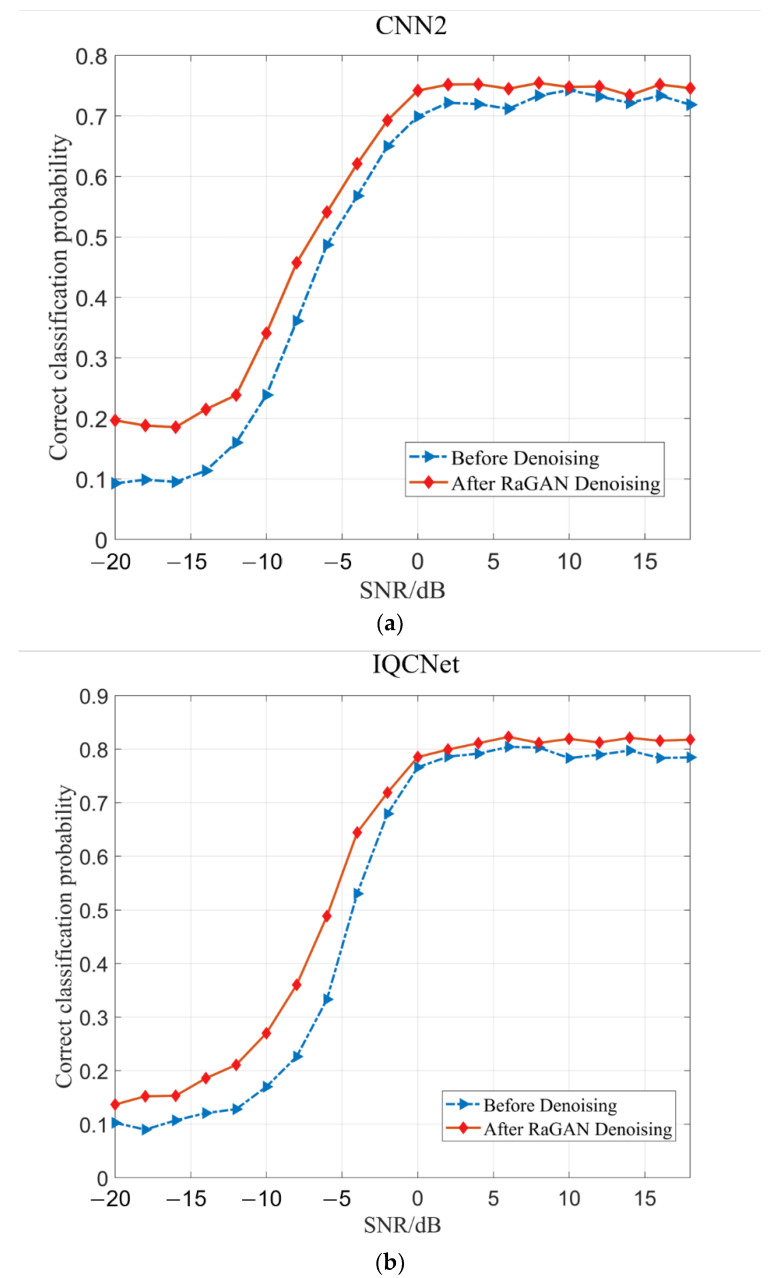
Comparison of recognition rate before and after RaGAN noise reduction: (**a**) CNN2 recognition comparison; (**b**) IQCNet recognition comparison; (**c**) IQCLNet recognition comparison.

**Figure 8 sensors-23-00475-f008:**
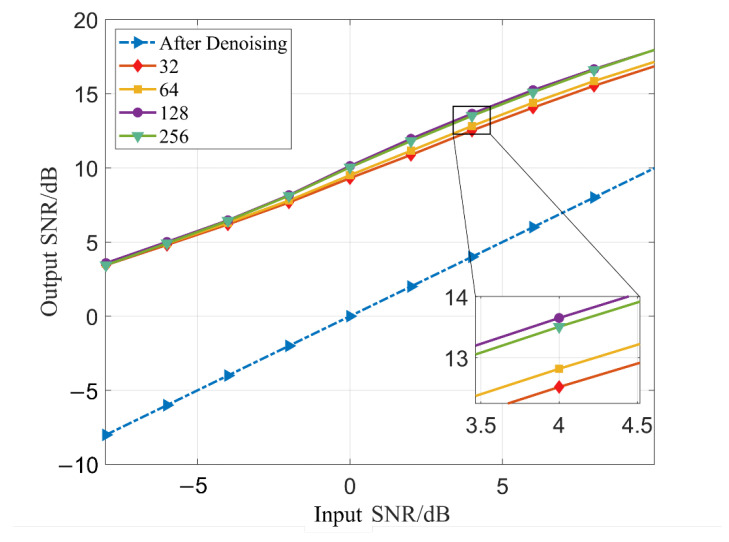
Comparison of noise reduction performance of different number of hidden nodes.

**Table 1 sensors-23-00475-t001:** List of abbreviations.

Abbreviation	Definition
DL	Deep learning
RaGAN	Relativistic average generative adversarial network
Bi-LSTM	Bidirectional long short-term memory
SNR	Signal-to-noise ratio
WT	Wavelet transform
EMD	Empirical mode decomposition
GAN	Generative adversarial network
AWGN	Additive white Gaussian noise
CNN	Convolutional neural networks
ECG	Electrocardiogram
LSGAN	Least-square generative adversarial networks
CGAN	Conditional generative adversarial networks
TEM	Transient electromagnetic
ANN	Artificial neural network
RF	Radio frequency
D	Discriminator
G	Generator
LSTM	Long short-term memory
LN	Layer normalization

**Table 2 sensors-23-00475-t002:** Dataset details.

Dataset Information	
Modulation	8PSK, BPSK, CPFSK, GFSK, PAM4, 16QAM, 64QAM, and QPSK
Length per Sample	128
Sampling Frequency	1 MHz
SNR Range	[−8 dB, −6 dB,…, 10 dB]
Total Number of Samples	80,000 vectors

**Table 3 sensors-23-00475-t003:** Configuration of hardware and software.

Hardware or Software	Technical Parameter
Operation System	Windows 10 Home Chinese
CPU	Intel Xeon Silver 4212R
GPU	NVIDIA GeForce 3090
Memory	128 G
Python	Python 3.8.12
Pytorch	Pytorch 1.11.0

**Table 4 sensors-23-00475-t004:** Hyperparameter of model.

Hyperparameter	
λ	0.005
Batch size	2048
Epochs	100
Learning rate	0.001
Optimizer	Adam [31]

**Table 5 sensors-23-00475-t005:** Specific data of noise reduction performance comparison of different methods.

SNR/dB	RaGAN (dB)	GAN (dB)	WT (dB)	EMD (dB)
−8	3.58	3.30	0.83	−0.16
−6	5.01	4.73	2.13	1.46
−4	6.47	6.21	3.55	3.31
−2	8.16	7.85	5.00	4.45
0	10.12	9.64	6.65	6.41
2	11.96	11.33	8.25	6.21
4	13.65	13.01	9.96	8.07
6	15.24	14.58	11.61	10.14
8	16.65	16.08	13.39	12.07
10	17.96	17.38	15.19	13.96

**Table 6 sensors-23-00475-t006:** Hyperparameter of the classification network model.

Hyperparameter	
Training set:Test set:Verification set	6:2:2
Batch size	1024
Epochs	32
Learning rate	0.001
Optimizer	Adam

**Table 7 sensors-23-00475-t007:** Comparison of time spent in processing 1024 noisy signals.

	WT	EMD	RaGAN
Time (s)	2.113	9.848	0.767

**Table 8 sensors-23-00475-t008:** Specific data of noise reduction performance Comparison of different number of hidden layer nodes.

SNR/dB	32 (dB)	64 (dB)	128 (dB)	256 (dB)
−8	3.44	3.53	3.58	3.44
−6	4.80	4.90	5.01	4.99
−4	6.19	6.30	6.47	6.43
−2	7.66	7.80	8.16	8.12
0	9.30	9.505	10.12	10.04
2	10.87	11.15	11.96	11.81
4	12.52	12.82	13.65	13.51
6	14.05	14.39	15.24	15.10
8	15.52	15.85	16.65	16.58
10	16.86	17.15	17.96	17.96

## Data Availability

Not applicable.

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
