# Peer review of "A Method of Noise Reduction for Radio Communication Signal Based on RaGAN"

_sensors, 2023, doi:10.3390/s23010475_

Round 1

Reviewer 1 Report

The authors have proposed a method to denoise the signal before detecting the modulation using DL technique and claimed to improve the signal modulation recognition by 10% (overall). The article is well written and well presented, however, there are few observations:

1- There are certain claims in the Section: Introduction. first paragraph that should include references.

2- The first section should highlight the shortcomings/research gap in the existing literature as well as novelty/contribution of the proposed solution. Also, the last paragraph usually mentions how rest of the article is organized.

3- There is typo in caption of figure 8

4- Since the authors have used public dataset and quite a few research studies have already used the same dataset, the state-of-the-art comparison may be presented to further highlight the achievements.

5- The complexity analysis must be added, since the data is for communication applications.

6- Authors have only used AWGN channel, it will be interesting to see how the proposed solution will work in Rayleigh fading.

Reviewer 2 Report

This paper tackles the problem of noise reduction of radio communication signal, The paper proposes a radio communication signal denoising method based on Relativistic average Generative Adversarial Networks which combines the Bi-directional Long Short-Term Memory model with the relative average generative adversarial network.

On the other hand, the paper should be revised by considering the following issues:

MAJOR ISSUES

+ Introduction section should be improved to give the motivation more clearly.

+ The main contributions of the paper should be clearly given as a separate subsection in the introduction section.

+ The organization of the paper should be clearly given as a separate subsection in the introduction section.

+ Especially considering the popularity of this problem, the related work should be improved. The number of references are insufficient. The related work and bibliography should be improved.

+ Most of the references in this paper are mostly recent publications (within the last 5 years) and relevant. On the other hand, the bibliography should be improved by adding most recent references.

+ The current “Related Work” section is in fact Background section with its mathematical details. Therefore, let's consider this section as "3. Background" section. “Related Work” section should be given as a separate section (as Section 2 before Section 3. Background).

+ Preamble information is required between section "2. Related Work" and subsection "2.1. Signal Denoising".

+ Section “Problem Definition and System Model” should be provided clearly as a separate section.

+ Preamble information is required between section "3. Signal Denoising Method Based on RaGAN" and subsection "3.1. Denoising Model".

+In Section "3.2. Dataset" takes SNR values from the set of [-8dB,-6dB,…,10dB]. What is the motivation choosing this values? They can come from the public dataset but still its motivation need to be given in the paper.

+"Dropout" and "Leaky ReLU" should be more emphasized in Figure 4. They are following "Fully connected" layers but it is hard to notice.

+What is the confrontation loss mentioned in Equation (7)? The confrontation loss should be give before Equation (7).

+"L_{MSE}" and "L_1" should be introduces before Equation (8).

+ Preamble information between section "4. Experiments" and subsection "4.1. Training Details and Parameters" should be improved.

+ The proposed scheme performs well. The motivation behind it should be explained better.

+ The figures/schemes are generally clear. They show the data properly. It is not difficult to interpret and understand them. On the other hand, Figure 1 should be explained better by adding more information to its caption and also give more details about some components like generator and discriminator in the figure.

+ Section "4. Experiments" should be improved. Figures should be clearly explained, especially in the text/main body of the paper.

+ The conclusion should be improved by giving the key results and main contributions more clearly.

+ Future work part should be given in the conclusion section.

 MINOR ISSUES

+ The grammatical errors and typos should be fixed.

+ Size of Figure 5, 6, 7, 8 should be increased.

+ Figure 2 should not exceed page margins so its size should be reduced.

+ The references in the bibliography should be given in the same style. The following link should be checked: https://www.mdpi.com/authors/references 

Round 2

Reviewer 2 Report

The paper is generally well written. The authors addressed my comments on the previous version of the paper considerably. On the other hand, they should consider the following issues:

+ Wide gaps at the bottom of Page 9 and 10 should not be left.

+ Minor spell-check is required.

+ Legends should be given in Figure 5.

+ The authors considers AWGN as model and use GAN to extract features; however, GAN may exhibit bad performances in some cases as mentioned in the following paper on RF Fingerprinting.

Ceren Comert, Michel Kulhandjian, Omer Melih Gul, Azzedine Touazi, Cliff Ellement, Burak Kantarci, and Claude D'Amours. 2022. Analysis of Augmentation Methods for RF Fingerprinting under Impaired Channels. In Proceedings of the 2022 ACM Workshop on Wireless Security and Machine Learning (WiseML '22). Association for Computing Machinery, New York, NY, USA, 3–8. https://doi.org/10.1145/3522783.3529518

The authors should provide their motivation for why they consider AWGN as their channel model. Then, they should present the motivation and the performance of GAN-based approach by considering their low-performance case given in the abovementioned paper.
